# Optimizing Hydrolysis Resistance and Dispersion Characteristics via Surface Modification of Aluminum Nitride Powder Coated with PVP-*b*-P(St-*alt*-ITA) Copolymer

**DOI:** 10.3390/molecules27082457

**Published:** 2022-04-11

**Authors:** Yu Wang, Shun Wang, Guangdong Zhu, Jianjun Xie, Zhan Chen, Ying Shi

**Affiliations:** 1Department of Electronics and Information Materials, School of Materials Science and Engineering, Shanghai University, Shanghai 200444, China; wangliuliu@shu.edu.cn (S.W.); xiejianjun@shu.edu.cn (J.X.); 2Shanghai Yuking Water Soluble Material Tech Co., Ltd., Shanghai 201318, China; sevenz@unipolymer.com (G.Z.); chenzhn621@unipolymer.com (Z.C.)

**Keywords:** RAFT polymerization, poly(vinyl pyrrolidone), block copolymer, AIN powder, surface modification, hydrolysis resistance, dispersion

## Abstract

Developing new coating modification technology of aluminum nitride (AlN) powder for higher hydrolysis resistance is the key to prepare high-performance AlN ceramic substrate with water-based wet process in the future. In the this paper, The poly(vinyl pyrrolidone)-*b*-poly(Styrene/Itaconic anhydride) (PVP-*b*-P(St/ITA))block copolymer with PVP as the independent chain segment was designed and synthesized through reversible addition fragmentation chain transfer (RAFT) polymerization, which was used for the study on coating modification, hydrolysis resistance, and dispersion performance of AIN powder. The study results show that, when using PVP macromolecular chain transfer agent (PVP-CTA) for the RAFT chain extension and polymerization in St/ITA binary system, the molecular weight increases linearly and the molecular weight distribution tends to decrease with the monomer conversion rate, which is in line with the activity-controlled characteristics of RAFT polymerization. The copolymer PVP-*b*-P(St/ITA) was used to for surface modification treatment of submicron AlN powder to generate esterification reaction, which was absorbed and bound to the powder surface. Hydrolysis resistance and dispersion experiments were conducted for modified powder, and the crystal phase and micro structure of modified powder were analyzed and observed through XRD, SEM, and TEM. It was found that copolymer modification had no effect on the powder crystal phase. A 8–21 nm passivation layer was coated on the surface, which can exist stably for 10 h in 60 °C water. Zeta potential and laser particle analyzer tests showed that modified powder featured excellent water-based slurry dispersion performance, and certain self-dispersing characteristics. The highest Zeta potential appeared in pH 6~7, and the particle granularity was distributed uniformly with the median particle diameter of 875 nm. The powder hydrolysis resistance and dispersion performance are significantly improved.

## 1. Introduction

Modern power devices impose high requirements on heat dissipation during work. Therefore, AlNceramic substrate material that matches with the Si semiconductor material is gradually becoming the ideal material for heat-dissipated package during the work of modern power devices thanks to its high heat conductivity, dielectric constant, and coefficient of thermal expansion [1,2,3]. Contributing to greener process makes water-based tape casting one of most promising development directions in ceramic forming process. Ceramic powder is the precursor for preparing ceramic substrate. Its purity, particle size, and uniform and stable dispersion in water-based slurry play a key role in the preparation of high-performance ceramic substrates with water-based tape casting [4]. Currently, ultra-fined and nano-sized ceramic powder is the synthetic trend of high-performance powder, but it poses such problems as higher surface energy of powder and easy conglomeration [5], which put forward new requirements for the preparation dispersion of tape casting slurry.

Compared with organic tape casting, when water serves as the solution in water-based tape casting, it provides AIN powder with additional hydroxyl or coordinated water molecules. Then, the powder is easily condensed through hydroxyl or coordinated water molecules and results in hard aggregation of powder. Meanwhile, during the preparation of AlN ceramic substrates with water-based tape casting technology, another serious problem is that the AIN powder for preparing ceramic substrate is prone to be hydrolyzed and oxidized to AlOOH and Al(OH)_3_, thus increasing oxygen content of powder [6,7]. During actual application, coexistence of easy hydrolysis and agglomeration in AIN powder significantly affects the powder purity and dispersion stability in water-based slurry, and further results in poor sintering and ceramic performance. Therefore, appropriate treatment for AIN powder is required.

AIN powder surface modification is the main channels to improve its hydrolysis resistance effect. There are two main methods: chemical modification and physical coating modification [8,9,10,11,12]. Wrapping the powder surface or forming a thin reaction layer can prevent interaction between AIN powder and water to improve its hydrolysis resistance capacity. Currently, AIN powder surface modification, regardless of small molecule inorganic acid modification, coupling agent modification, or macromolecular polymer modification, is based on its hydrolysis resistance effect without further consideration of powder dispersion performance. In particular, during the preparation of water-based slurry, high-energy treatment such as ball billing is required for good dispersion effect in slurry. However, after such high-energy mechanical treatment, the coating layer is easily damaged, degrading hydrolysis resistance of powder [13]. Therefore, to ensure good dispersion performance of modified powder, reducing the dependence on high-energy mechanical dispersion is paramount to the implementation of water-based wet forming preparation process of AlN ceramic substrate.

Poly(vinyl pyrrolidone) (PVP) is the homopolymer of N-vinyl pyrrolidone (VP). Thanks to the strong dipolarity of its lactam group, it exhibits good absorption capacity for solid particles, and is widely applied in powder dispersion, in particular preparation dispersion of nanometer powders. Application fields include carbon nano tubes, nano silver wire, etc. [14,15,16]. However, due to a lack of active groups, it is difficult for the PVP structure to have a bonding effect with powder surface and to form stable chemical bonds. Therefore, by introducing other monomers with specific functional group in PVP structure to form VP copolymers with certain structure features, on the one hand, the inherent dispersion performance of PVP can be maintained. On the other hand, esterification reaction can be generated between other specific functional groups of monomer and the active site hydroxy group (–OH) on the AIN powder surface to form a passivation layer. This makes it possible to improve the AIN powder’s application performance in water-based tape casting from two aspects of dispersion and hydrolysis resistance. In addition, in subsequent ceramic high-temperature sintering process, VP copolymer can be decomposed to produce volatile gas such as CO_2_, and causes no adverse effect on the heat-conducting property of AlN ceramic.

Itaconic anhydride (ITA) is an important itaconic acid derivative and fine chemical raw material, and is renewable resources [17]. Containing active functional groups like unsaturated double bond and anhydride in its molecules, it is easy to generate copolymerization with other monomers. In addition, high-active anhydride group also provides copolymer with rich derivative reaction characteristics. Introducing ITA into the PVP structure not only provides PVP with new active site of reaction, but also has a positive significance for the development application of itaconic acid series derivatives.

In this paper, based on the AIN powder water-based dispersion and hydrolysis resistance demands, a block copolymer with PVP as the independent segment was designed and synthesized. First, the hydroxyl-terminated PVP (PVP-OH) was prepared using isopropyl alcohol (IPA) and β-mercaptoethanol (MCE) as the chain transfer agent. Then, esterification condensation reaction was generated between RAFT reagent with carboxyl (-COOH) as the terminal group and PVP-OH to form macromolecular chain transfer agent PVP-CTA. PVP-*b*-P (St/ITA) block copolymer was designed and synthesized through the binary system RAFT chain extension reaction between PVP-CTA and St/ITA. Then, a characterization analysis was conducted for the structure and molecular weight through FT-IR, ^1^H-NMR, and GPC. Then, after coating modification with PVP-*b*-P(St/ITA) copolymer on AlN surface, the crystal phase and micro structure of modified powder were analyzed and observed through XRD, SEM, and TEM. It was found that copolymer modification had no effect on the powder crystal phase. The AIN powder surface features a 8–21 nm thick uniform passivation layer. With concentrated distribution of grain diameters in the water-based system, the powder has certain self-dispersing characteristics with the median particle diameter of D_50_ less than 900 nm, and average particle size closed to the original particle size. After coating modification, AIN powder remains stable in water-based suspension at 60 °C for 10 h without hydrolysis. 

## 2. Results

### 2.1. Synthesis of Hydroxyl-Terminated PVP-OH

In this paper, hydroxyl-terminated PVP-OH was synthesized first, and esterification reaction was generated between PVP-OH and the carboxylic acid anhydride RAFT reagent to synthesize macromolecular chain transfer agent PVP-CTA. Then, a RAFT chain extension experiment was conducted for monomer with specific functional groups. Figure 1 shows the synthesis process. It can be seen that, in the synthesis of PVP-OH, MCE serves as the chain transfer agent, and IPA serves as the solvent. Under certain reaction conditions, the VP terminal group tends to generate chain transfer reaction between MCE and IPA, so as to generate hydroxyl-terminated functional PVP-OH. By changing the ratio between MCE, IPA, and VP, it can effectively adjust the synthetic PVP-OH molecular weight indicators. However, due to extremely high MCE chain transfer constant, at the time of one-time adding during feeding stage, most MCE has generated chain transfer reaction in the initial stage of reaction (i.e., VP oligomer stage) [18]. This is not conducive to the PVP-OH molecular weight adjustment and molecular weight distribution. The experiment adopted the dropwise adding of MCE for PVP-OH preparation to control MCE within a certain concentration in the medium and later stages of polymerization. Table 1 shows the effect of different mercaptan feed ratios and dropwise adding times on the PVP-OH molecular weight and molecular weight distribution. It can be observed that, compared with one-time adding of MCE, dropwise adding MCE can effectively reduce the PVP-OH molecular weight. The main reason is that compared with the initial one-time adding where MCE is consumed up in the VP oligomer stage, dropwise adding method can ensure a certain MCE concentration during the chain propagation step of polymerization, and enhances the chain transfer reaction with VP polymers with medium and high-molecular weights, thus lowering molecular weight. Regarding different dropwise adding times, as the dropwise adding experiment extends, molecular weight tends to decrease. However, the molecular weight distribution tends to decrease and then increase. The main reason lies in that the growth stage of free radical polymerization is divided into an initial stage, middle stage, and late stage. In the initial stage, the monomer conversion rate is in approximate linear relationship with time. The middle stage is the acceleration stage, while the late stage is the deceleration stage. Most monomers finish polymerization in the middle stage [19]. Hence, controlling the MCE dropwise adding time period within the middle period can effectively control the product molecular weight and molecular weight distribution. When dropwise adding time is extended to the chain’s late stage, the unpolymerized monomer concentration is obviously reduced, and MCE adding will only result in obvious production of VP oligomer and enlarging molecular weight distribution.

### 2.2. Synthesis of Block Copolymer PVP-b-P (St/ITA)

This experiment adopted the chain transfer agent method to adjust the MCE adding method and synthesize terminated PVP-OH with low molecular weight and molecular weight distribution. Selected PVP-OH (Mn = 2100) to conduct condensation esterification with RAFT reagent DAAC to synthesize macromolecular chain transfer agent PVP-CTA. Then, PVP-CTA was used as the macromolecular chain transfer agent for St/ITA binary RAFT chain extension copolymerization, in which the molar ration between St and ITA is 1:1. Figure 2a shows the GPC elution time curve of chain extension product at different monomer conversion rates. It can be seen that, as the monomer conversion rate increases, the product molecular weight increases and appears single-peak distribution. This indicates that PVP-CTA has effectively controlled St/ITA for chain extension reaction, but not the monomer self-polymerization reaction. Figure 2b shows the molecular weight and molecular weight distribution of chain extension product at different monomer conversion rates. The molecular weight generally appears a linear increase trend with the increase of monomer conversion rate, and molecular weight distribution appears a general decrease trend as molecular weight increases. It can be seen that, PVP-CTA has good molecular weight activity control effect on St/ITA binary system, which is in line with the RAFT polymerization reaction characteristics.

The polymerization characteristics of itaconic anhydride are similar to maleic anhydride. Due to large steric hindrance, it is hard to generate polymerization. However, it is a strong electron acceptor, and is easy to form charge transfer complex (CTC) with strong electron donor, such as styrene [20,21,22], as shown in Figure 3.

The experiment tested the N content and acid value of samples at different conversion rates in the polymerization process to determine the molar ratio between St and ITA in the chain extension product, as shown in Table 2. It can be seen that the molar ratio between St and ITA in chain extension product is basically maintained at 1:1 at different conversion rates. The molar ratio of them in the copolymer is approximate, and basically maintains an alternating copolymerization trend. Hence, the block copolymer results can be inferred as PVP_19_-*b*-P(St_31_-*alt*-ITA_31_).

### 2.3. FT-IR Characteristic

Figure 4 shows the infrared spectrogram of PVP-CTA and PVP-b-P (St/ITA). In the figure, absorption peak 1679 cm^−1^ is the C=O stretching vibration absorption peak in PVP Lactam group, 1286 cm^−1^ is the C-N stretching vibration absorption peak in lactam, and 1732 is the ester base peak after the condensation of PVP-OH and DDAC, which certifies successful synthesis of PVP-CTA. Compared with PVP-CTA, in the infrared spectrogram of chain extension product PVP-*b*-P (St/ITA), 3060 cm^−1^ and 3030 cm^−1^ are the CH stretching vibration absorption of benzene ring; 750 cm^−1^ and 703 cm^−1^ are the out-of-plane bending vibration absorption of benzene ring, certifying the existence of the styrene structure. The absorption peak 1850 cm^−1^ and 1780 cm^−1^ are the C=O symmetric and asymmetric stretching vibration absorption of itaconic anhydride, and 1224 cm^−1^ is the C-O stretching vibration absorption of the pentacyclic anhydride, certifying the existence of itaconic anhydride in polymer. Combining the product’s purification process and the abovementioned polymer characteristic absorption peak, it indicates the successful synthesis of PVP-*b*-P (St/ITA) block copolymer.

### 2.4. H-NMR Characteristic

Figure 5 is the ^1^H-NMR (δ, ppm, DMSO) of PVP-b-P (St/ITA) block copolymer, and δ = 2.50 is the solvent DMSO solvent peak. 6.7~7.4 is the characteristic chemical shift of hydrogen of benzene ring (c) in the St structure; 3.46~3.88 is the characteristic chemical shift of hydrogen on hypomethyl (a) in the NVP main chain; 3.27~3.46 is the characteristic chemical shift of the anhydride ring (d) in the ITA; 2.92~3.26 is the characteristic chemical shift of the hydrogen on methylene (b) connected to N in the pyrrolidone heterocycle of NVP; 2.17~2.46 is the characteristic chemical shift of the methyne (e) in the St main chain; 1.97~2.17 is the characteristic chemical shift of the hydrogen on methylene (f) linked with carbonyl in the pyrrolidone heterocycles of NVP; 1.74~1.97 is the combined characteristic chemical shift of hydrogen on methylene (j, i) in the NVP, St main chain; 1.41~1.74 is the combined characteristic peak of the methylene (h) and pyrrolidone heterocycle methylene (g) in the ITA main chain; 1.32 is the characteristic chemical shift of DDAC methylene (k) of the RAFT reagent; and 1.0 is the combined peak of the PVP isopropyl and methyl (p) in DDCA. The related hydrogen spectrum chemical shift of polymerization product is embodied in the spectrogram. Combined with the FT-IR analysis results, it can be determined that the final synthetic product is the PMA-*b*-P (St/ITA) block copolymer product.

### 2.5. Characteristics of Modified AlN Powder Coated with PVP-b-P (St/ITA) Copolymer

#### 2.5.1. Hydrolysis Test of Modfied AlN

AlN powder is easily eroded in water to generate Al (OH)_3_ and NH_3_ by hydrolysis, resulting in rising pH of solution. Therefore, observation of pH change of AlN powder in the aqueous suspension is one of the important indicators to measure the hydrolysis resistance of modified AlN powder coated with copolymer. Added 2 g raw AlN powder (R-AlN) and modified AlN powder (P-AlN) with PVP-*b*-P (St/ITA) block copolymer into 50 mL deionized water and stirred it uniformly until suspension liquid appeared. Placed the solution in a 60 °C oven for heat preservation. Monitored the change of pH with time of the suspension system, as shown in Figure 6. The figure shows that for powder before modification treatment, the pH value within 1 h in the 60 °C water bath reaches 10, indicating that the hydrolysis is basically completed. For samples after copolymer modification treatment, the pH value basically maintains stable at 4~5 within 10 h. Compared with original powder, modified powder exhibits good hydrolysis resistance even in high temperature. After 10 h, pH shows a clear increasing trend. Under Brownian movement, hydrones diffuse over copolymer and form a coating layer on the powder surface, which penetrates into the AlN powder surface to generate hydrolysis.

XRD spectrum can be used for the phase analysis before and after AlN powder hydrolysis, and is an important method to characterize the AlN crystal phase change. Conduct XRD characterization for R-AlN and P-AlN powder soaked in water-based medium with different times to further investigate the change of phase, as shown in Figure 7. It can be found that, the phase diffraction peak of R-AlN has completely changed after 1 h. Based on PDF standard card analysis, main hydrolysis products are Al(OH)_3_ and AlOOH. However, P-AlN powder is still essentially the same as the AlN standard diffraction peak, without other obvious diffraction peaks. This indicates that the modification of AlN powder using PVP-*b*-P (St/ITA) block copolymer has no impact on the AlN phase lattice, and the modification treatment only happens on the powder surface. In addition, within 10 h in 60 °C, modified powder shows no impurity peak, and its peak pattern is the same as the original powder, without any hydrolysis phenomenon, but maintains AlN phase characteristics. The AlN powder after block copolymer modification displays good hydrolysis resistance performance.

#### 2.5.2. SEM and TEM Analysis

Figure 8 is the SEM image of AlN powder. In the figure, Figure 8a shows SEM of original AlN powder, which appears irregular smooth spheroid, and particles are relatively aggregated closely to each other. The original particle size about 0.6 μm. Figure 8c is the SEM of AlN powder after 1 h of hydrolysis, which shows long-rod structure or block structure, in which, the entire structure becomes larger, and the shape significantly changes. Figure 8b is the SEM of AlN powder after block copolymer modification, which is essentially the same as the original AlN, in which, the surface is relatively smooth in an irregular ball shape, and the powder dispersion is relatively even without obvious agglomeration. Figure 8d shows the modified AlN powder after 10 h of hydrolysis. The agglomeration is not significantly changed, remaining in a regular spheroid and showing good dispersion state between powder particles. It can be observed that the copolymer-modified powder exhibits notable improvement in hydrolysis resistance and dispersion effect.

The AlN/copolymer core-shell structure can be clearly observed on the surface of modified AlN powder under a transmission electron microscope. It is found that the surface is covered with one macromolecule coating layer of about 8–21 nm, and the block copolymer has been effectively bound on the powder surface, as shown in Figure 9.

#### 2.5.3. Zeta Potential and Particle Size Analysis

When studying powder dispersion system, it is generally necessary to measure the change curve of powder Zeta potential with slurry pH, whose value is related to the powder dispersion stability. Zeta potential is an important measure of characterizing powder particle dispersion stability, and also an important index for the attraction and repulsion strength between powder particles. Disperse 1 g AlN powder with ultrasonic dispersion in 50 mL water and adjust different pH to determine Zeta potential, as shown in the Figure 10. When original AlN powder pH is <9–10, it is easy to absorb H^+^, thus carrying positive charge, and therefore the zeta potential is positive; when pH is >9–10, it is easy to absorb OH^−^, thus carrying negative charge, and therefore, the zeta potential is negative. When the isoelectric point (IEP) in the aqueous solution is between pH = 9–10, it is most likely to be precipitated, and the slurry is unstable. For the modified AlN powder, the Zeta potential in low pH phase (pH < 5) is less than original powder. This is mainly due to the fact that, different from ion adsorption, in the AlN powder coated with high molecule, the high molecule has steric hindrance on the particle surface so that the shearing surface moves away from the particle surface. As a result, the Zeta potential is decreased. In addition, the PVP lactam group and ring-opening carboxyl group of anhydrides are prone to generate electrostatic charge neutralization with H^+^ in acidic condition conditions, thus decreasing Zeta potential. When the pH value is 6~7, the modified powder exhibits the highest Zeta potential value, which is about 50 mV. This is largely due to the basic ionization of carboxylic acid groups in the copolymer structure, so the chain segment fully extends to obtain large electrostatic repulsion. As pH value further increases, Zeta potential tends to further decrease. The main reason is that increasing negative charges in the solution compress the extension of the carboxylic acid groups in copolymer, thus reducing the electrostatic repulsive force from double electrode layer and powder solution stability. Controlling powder slurry pH within 6~7 can obtain better dispersion effect for modified AlN powder.

The dispersed particle size distribution of AlN powder in water-based slurry is an important factor that affects later ceramic sintering performance. Take an appropriate amount of raw and modified AlN powder for ultrasonic dispersion in water, adjust the R-AlN and P-AlN solution pH to 3~4 and 6~7, and test their particle sizes, as shown in Table 3 and Figure 11 below. Notable improvement in particle size distribution is observed in the modified powder, and the overall particle size distribution is markedly decreased. The average particle size is more closed to the original particle size, and the dispersion is more evenly concentrated. The copolymer’s surface modification for AlN powder not only realizes the passivation protection of powder surface, but also inhibits the agglomeration characteristics of powder under water-based condition and achieves certain self-dispersing capacity, which exhibits goods dispersion characteristics without high-energy mechanical treatment. This can effectively reduce the dependence on mechanical dispersion and enable greener process of water-based wet forming.

## 3. Materials and Methods

### 3.1. Reagents and Instruments

N-vinyl pyrrolidone (VP,99.5%, Shanghai Yuking Tech, Shanghai, China); styrene (St, 99%, Shanghai Macklin, Shanghai, China) after distillation under reduced pressure and purification; itaconic anhydride (ITA, 99.5%, Shanghai Macklin, Shanghai, China); 2,2′-azobis[2-methyl-n-(2-hydroxyethyl)propionamide) (VA-086), azodiisobutyronitrile (AIBN),β-mercaptoethanol (β-MCE), isopropyl alcohol (IPA), N,N′-Dicyclohexylcarbodiimide (DCC), 4-Dimethylaminopyridine (DMAP), acetone, dioxane, 1-dodecanethiol, carbon disulfide, 2-bromine propionic acid, sodium hydroxide, ethyl alcohol, AlN powder (original particle size: approximate 600 nm) were analytical reagents, from Shanghai Macklin Biochemical Co., Ltd. (Shanghai, China).

### 3.2. Characterization of the Copolymer

FT-IR: The structure of copolymer was characterized by FT-IR (KBr, Shimadzu Fourier transform infrared spectrometer IRSpirit-T (Shimadzu Corporation, Kyoto, Japan), and the range of scanning is between 4000–500 cm^−1^.

^1^H-NMR: The ^1^H-NMR of copolymer was characterized by Bruker Advance spectrometer (Bruker Corporation, Billerica, MA, USA) at 400 MHz, and DMSO as solvent.

GPC: GPC was used to determine the molecular weight and polymer dispersion index of all polymers used in this work, and Shimadzu GPC (Shimadzu Corporation, Kyoto, Japan) was employed for analysis. The copolymer was neutralized and dissolved in an appropriate amount of sodium hydroxide solution in a water bath of 60–70 °C, with pH adjusted to 7–9, and filtered through a 0.45 μm filter membrane. The mobile phase employed 30% acetonitrile aqueous solution of 0.6% NaCl, flow rate of 0.8 mL/min, column temperature of 40 °C and mono-disperse PEG/PEO as reference standard to generate the calibration curve.

### 3.3. Experimental Process

#### 3.3.1. Synthesis of Chain Transfer Agent Didecyl Dimethyl Ammonium Chloride (DDAC)

According to the literature [23], in a nitrogen environment, 1-dodecanethiol (80.76 g, 0.40 mol), acetone (192.4 g, 3.31 mol) and tripropyl methyl ammonium chloride (6.49 g, 0.016 mol) were mixed in a flask, and cooled to 10 °C. 50% sodium hydroxide solution (33.54 g, 0.42 mol) was added dropwise, and the dropwise adding time was controlled over 20 min. After dropwise adding, continued stirring for 15 min, and dropwise added the mixture solution of carbon disulfide (30.42 g, 0.40 mol) and acetone (40.36 g, 0.69 mol); controlled the dropwise adding time over 20 min; during the process, the color of solution changed to red. Continued stirring for 10 min, and added 2-bromine propionic acid (59.8 g, 0.40 mol). Then, dropwise added 50% sodium hydroxide (160 g, 2 mol) solution, and stirred 30 min. Added 600 mL water, and then added 100 mL hydrochloric acid to acidize the aqueous solution. Inlet nitrogen and stirred violently to help acetone evaporation. Collected solid with Buchner funnel, and stirred it in isopropyl alcohol to filter undissolved solids. Concentrated isopropyl alcohol solution until it was dried, and recrystallized normal hexane to yellow crystalline solid. IR (KBr): 1732, 1076. ^1^H-NMR (δ, ppm, DMSO): 0.85 (t, 3H), 1.13–1.45 (m, 18H), 1.51 (d, 3H), 1.63 (m, 2H), 3.37 (t, 2H), 4.67 (q, 1H), 13.17 (s, 1H).

#### 3.3.2. Synthesis of Hydroxyl-Terminated PVP (PVP-OH)

Dissolved NVP (10 g, 90 mmol) and VA-086 (259 mg, 0.9 mmoL) in 150 mL isopropyl alcohol, added the solution to a three-necked bottle reflux unit, and emptied nitrogen for replacement for 15 min. After that, quickly heated the solution to 85 °C, and dropwise added β-MCE (176 mg, 2.25 mmoL) isopropyl alcohol solution. After dropwise adding, continued reflux reaction for 12 h [18]. Removed most isopropyl alcohol through decompression spin steaming, dissolved remnant with 20 mL dichloromethane, and precipitated the remnant in massive diethyl ether for three times. Vacuum dried the product at 60 °C overnight to obtain hydroxyl-terminated PVP-OH. Adopted GPC to test molecular weight (*M*n = 2100, *M*w/*M*n = 1.46).

#### 3.3.3. Synthesis of Macromolecular Chain Transfer Agent PVP-CTA

Added PVP-OH (5.25 g, 2.5 mmol), DDAC (1.12 g, 3.2 mmol), and a catalytic amount of DMAP to 30 mL DCM, and stirred them until even dissolution. In the ice-water bath, dropwise added the solution in DCC (1.03 g, 5 mmol)/DCM(20 mL) solution, and controlled the dropwise adding time over 30 min. After dropwise adding, stirred the solution in 40 °C for reaction of 24 h. Filtered and removed dicyclohexylurea and frozen the filter liquor overnight, and removed precipitate by filtration again. Removed some DCM through rotary evaporateion, and precipitated it in appropriate normal hexane, the pure product was obtained by repeated dissolution-precipitation operations for 2–3 times; vacuum dried the product in 60 °C overnight to obtain macromolecular chain transfer agent PVP-CTA. ^1^H-NMR (δ, ppm, DMSO): 3.44~3.85 (-C**H**N-), 3.37 (-SC**H**_2_), 2.92~3.26 (-NC**H**_2_-), 1.97~2.17 (-COC**H**_2_-), 1.73~1.96 (-CHC**H**_2_-), 1.42~1.73 (-C**H**_2_CH_2_CO-), 1.13~1.40 (-C**H**_2_-), 0.91~1.12 (-C(C**H**_3_)C**H**_3_), 0.8~0.88 (-CH_3_). 

#### 3.3.4. Synthesis of PVP-*b*-P (St/ITA)

Under nitrogen protection, added PVP-CTA (2.45 g, 1 mmol), AIBN 0.033 g (0.02 mmol), St (3.64 g, 35 mmol), ITA (3.92 g, 35 mmol), and 10 mL dioxane in sequence in a flask. After emptying nitrogen for protection for 30 min, raised oil bath temperature to 60 °C for reaction of 12 h. Cooled it in the ice-water bath and terminated reaction. Precipitated the product in ethyl alcohol. After DCM dissolving, dissolved and precipitated repeatedly for 2~3 times. Then washed it with hot hexane for several times. Conducted centrifugal separation and dried it in a vacuum drying oven at 60 °C to obtain block copolymer PVP-b-P (St/ITA). 

### 3.4. Effect Verification of AlN Powder Surface Modification

#### 3.4.1. Modification Treatment

Added 10 g AlN powder and 1 g block copolymer in 50 mL ethyl alcohol. After high-speed shear dispersion (2000 r/min) for 15~20 min, conducted ultrasonic treatment for 5~10 min. After full scattering and dispersion of AlN powder, stirred it in the 80 °C oil bath for modification treatment for 6 h. Removed it for filtration under diminished pressure. Taking down, centrifuged at 2000 r/min for 3 min, washed the dispersed precipitate with ethanol, centrifuged again, repeated treatment for 2–3 times, and vacuum dried at 50 °C.

#### 3.4.2. Hydrolysis Resistance and Dispersion Effect Experiments

Added 2 g modified AlN powder in 50 mL water for ultrasonic dispersion treatment for 2 min. After homogeneous dispersion, conducted a hydrolysis test in the 60 °C oven; recorded the dispersion liquid pH values in different time stages, and analyzed the hydrolysis resistance characteristics of modified powder using XRD (Japan, Rigaku, D/max-2550), SEM (South Korea, EM-30+, COXEM), TEM (Japan, JEOL, JEM F200), Zeta potential (UK, Malvern Instruments, Zetasizer 300HSA), and laser particle analyzer (China, Dandong Baxter, BT-90).

## 4. Conclusions

Based on the requirements for AlN powder dispersion and hydrolysis resistance, a binary RAFT chain extension reaction was conducted between the macromolecular chain transfer agent PVP-CTA and St/ITA, and the PVP-*b*-P(St/ITA) block copolymer with PVP as the independent chain segment was designed and synthesized for characterization using FT-IR, ^1^H-NMR, GPC, and other methods, indicating that the product is the target material. The molecular weight increases linearly with the monomer conversion rate, the molecular weight shows narrow distribution with activity-controlled polymerization characteristics, and the binary chain extension segment has a certain regular alternating structure.The prepared block copolymer is used for surface modification treatment of AlN powder, which effectively realizes binding coating on the powder surface, which maintains stably for over 10 h in 60 °C water and displays good hydrolysis resistance performance. Characterizing the modified powder using XRD, SEM, and TEM shows that the surface modification treatment has no effect on the AlN powder phase lattice. Copolymers are evenly coated on the powder surface to form an even passivation layer with thickness of 8–21 nm. After such tests as Zeta potential and laser particle analyzer, modified powder exhibits outstanding self-dispersing characteristic in water-based slurry. It has the highest Zeta potential of approximate 50 mV in pH 6~7, and the particle size distribution decreases significantly with more even dispersion. D_50_ is less than 900 nm, and the average particle size is close to the original particle size.

## Figures and Tables

**Figure 1 molecules-27-02457-f001:**
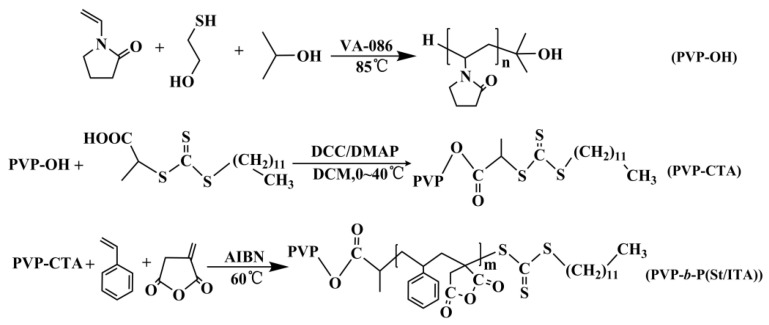
The synthesis process of PVP-*b*-P(St/ITA).

**Figure 2 molecules-27-02457-f002:**
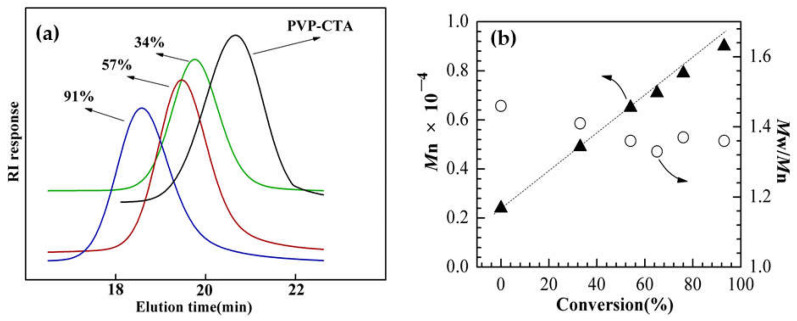
(**a**) shows the GPC elution curve of copolymer at different conversion rates; (**b**) shows the molecular weight and molecular weight distribution at different monomer conversion rates.

**Figure 3 molecules-27-02457-f003:**
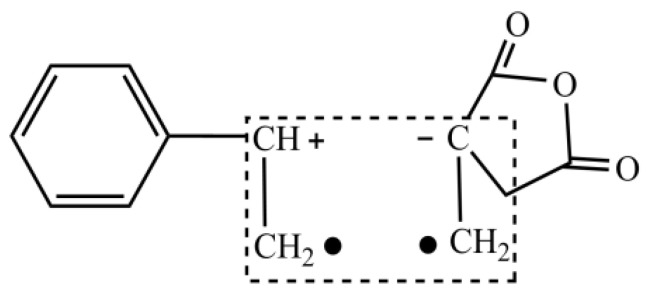
CTC Structure Formed by ITA and St.

**Figure 4 molecules-27-02457-f004:**
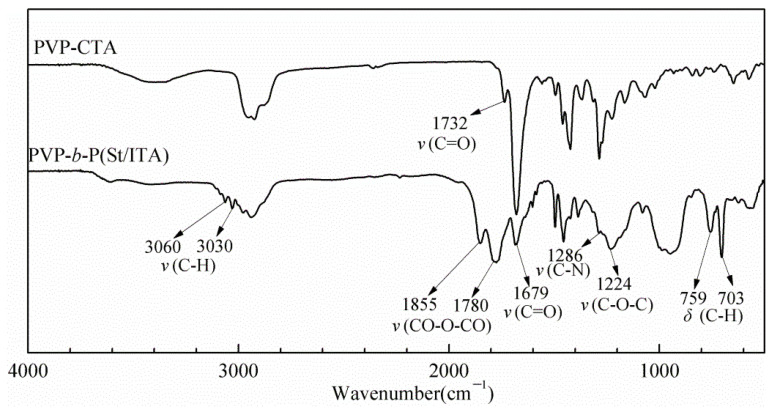
Infra-red spectrogram of PVP-CTA and PVP-*b*-P (St/ITA).

**Figure 5 molecules-27-02457-f005:**
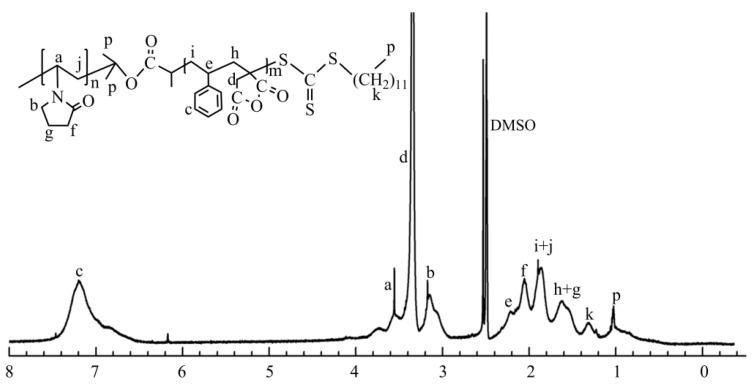
^1^H-NMR Spectrum of polymer PVP-*b*-P (St/ITA).

**Figure 6 molecules-27-02457-f006:**
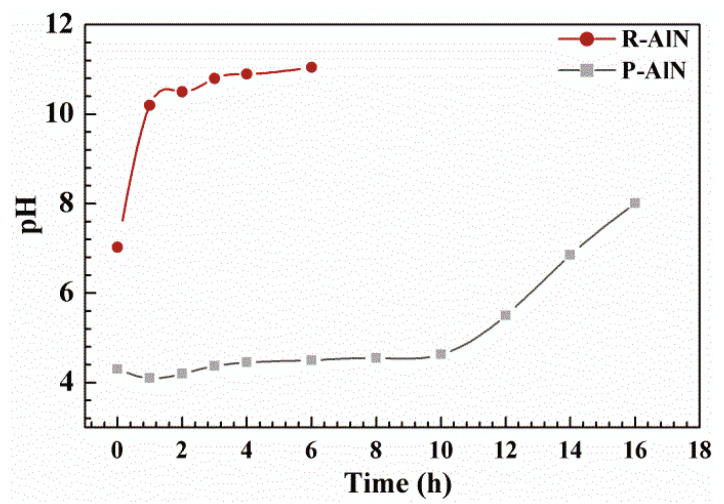
Change of R-AlN and P-AlN Powder pH value over time in 60 °C water-based medium.

**Figure 7 molecules-27-02457-f007:**
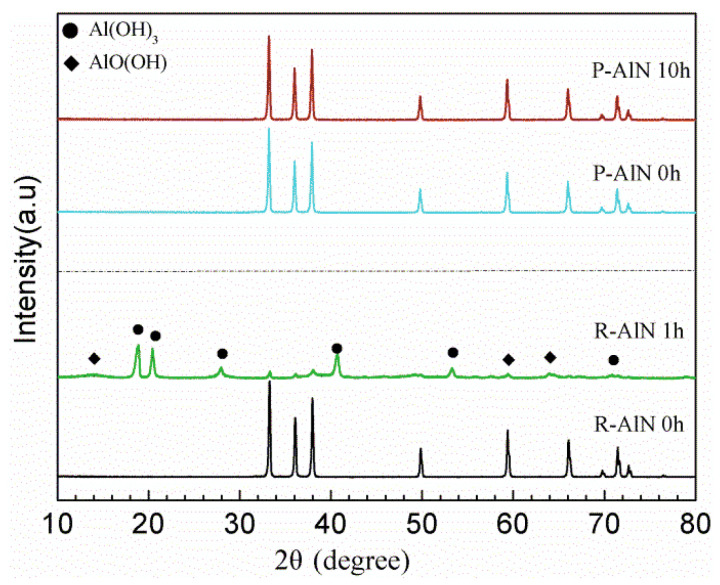
XRD spectrum of raw powder and modified powder under different hydrolysis resistance Times.

**Figure 8 molecules-27-02457-f008:**
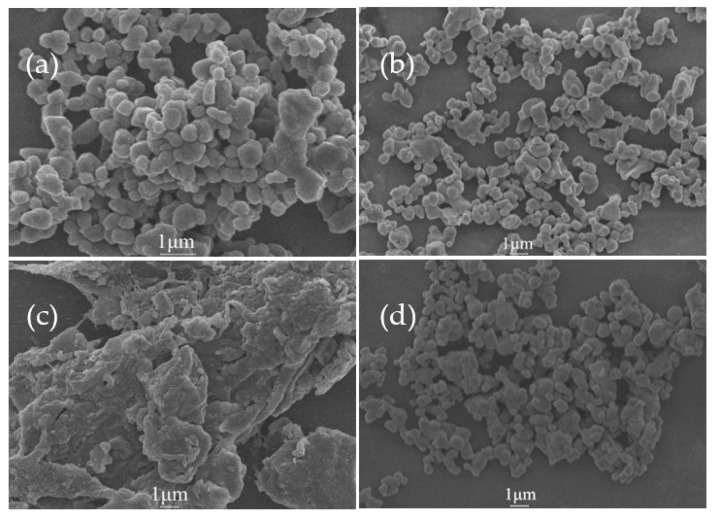
AlN powder SEM diagram: (**a**) original AlN powder; (**b**) modified AlN powder; (**c**) original AlN after 1 h of hydrolysis; (**d**) modified powder after 10 h of hydrolysis.

**Figure 9 molecules-27-02457-f009:**
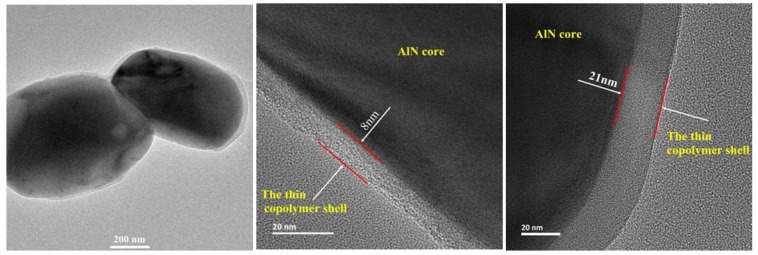
TEM diagram of modified powder.

**Figure 10 molecules-27-02457-f010:**
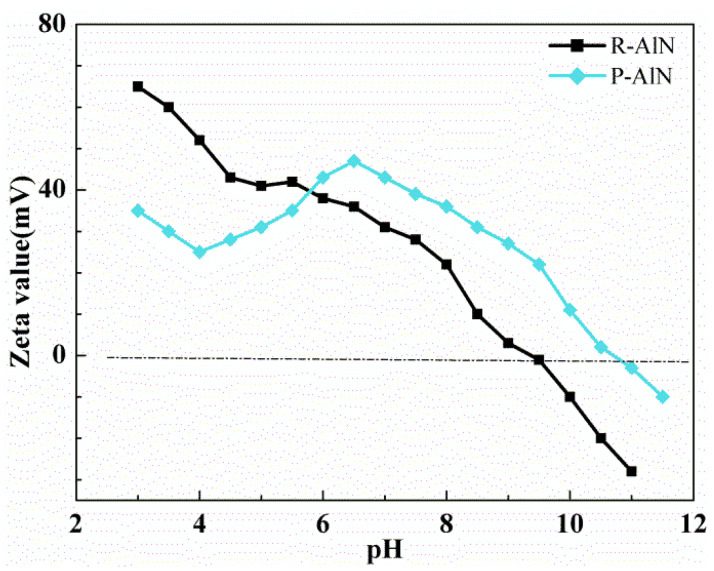
Zeta potential of AlN powder in different pH values.

**Figure 11 molecules-27-02457-f011:**
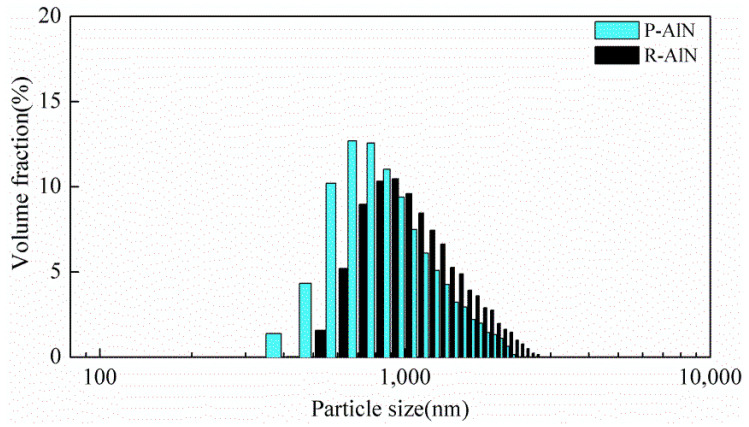
Particle size distribution diagram of AlN powder in water-based system.

**Table 1 molecules-27-02457-t001:** PVP-OH molecular weights with different mercaptans and dropwise adding times.

Group	VP (g)	MCE (mol%)	Adding Times (h)	*M*n^SEC^ (g mol^−1^)	Dispersity (*M*w/*M*n)
1	10	4	0	2400	1.62
2	10	4	1	2000	1.53
3	10	4	2	1700	1.36
4	10	4	4	1500	1.76
5	10	2.5	2	2100	1.46

**Table 2 molecules-27-02457-t002:** St/ITA molar ratio of chain extension product at different conversion rates.

Conversion	PVP-CTA	34% *	57%	76%	91%
N content/%	10.95	5.31	3.97	3.27	2.88
Acid value (mg/g)	-	273	335	358	371
St/ITA	-	0.96:1	0.97:1	1.03:1	1.06:1

* The mass ratio of different monomers in copolymer was determined by N content and acid value, and then the monomer conversion rate was calculated.

**Table 3 molecules-27-02457-t003:** Particle size distribution of AlN powder in water-based system.

Group	D_50_/nm *	D_75_/nm	D_90_/nm	Average Size (nm)
R-AlN	1149	1524	1929	1043
P-AlN	875	1189	1556	733

* D_X_, corresponding particle size when the cumulative percentage of grain size distribution of a sample reaches X%.

## Data Availability

Not applicable.

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
