# Peer review of "Optimizing Hydrolysis Resistance and Dispersion Characteristics via Surface Modification of Aluminum Nitride Powder Coated with PVP-b-P(St-alt-ITA) Copolymer"

_molecules, 2022, doi:10.3390/molecules27082457_

Round 1
Reviewer 1 Report
The aim of the study was to design a new technology of coating modification of aluminum nitride powder in order to obtain greater resistance to hydrolysis, which is the key to the future preparation of a high-performance aluminum nitride ceramic substrate with water-based wet process.
It's a well-written manuscript and the paper will be an interesting study that can contribute to the scope of the Molecules, however, the manuscript contains a lot of mistakes (some highlighted below), which should be corrected.
Authors should follow the SI rules when writing variables and their units (spaces, abbreviations for units, italics, subscripts, etc.). There are many punctuation errors (words written with a capital letter in the middle of a sentence, e.g. names of compounds or methods are not proper names).
The problem with the correct understanding of the text is made difficult by numerous abbreviations. Some of them are not explained at the beginning.
- Abbreviations in the title should be avoided (except for commonly used polymer names), so please use the whole name as AIN is not a commonly known abbreviation.
- Line 12, 14, 18, 97, 120: Please explain the abbreviation where it was first used.
- Line 26 and so on: Please use correct SI symbols. For hour or hours is h. Please correct throughout the text.
- Line 28, 70: “polyvinyl pyrrolidone”: According IUPAC nomenclature names of polymers whose monomers consist of two words or more are written with parentheses. Should be poly(vinyl pyrrolidone). Please correct.
- Line 51, 84: 3/2 should be subscript. Please correct.
- Table 1: “PDI” (polydispersity index);IUPAC has deprecated the use of the misleading term polydispersity index, having replaced it with the term “dispersity” for polymers, represented by the symbol ĐM. Please correct.
- Figure 5 and Line 243; Please explain the abbreviation “R-AlN and P-AlN”.
Reviewer 2 Report
The authors submitted the article entitled “Optimizing Hydrolysis Resistance and dispersion Characteristics via Surface Modification of AIN Powder Coated with PVP-b-P(St-alt-ITA) Copolymer”. I recommend that the paper could be accepted after major revisions. My main comments and questions are as follows:
1. In Tables, the authors should provide the footnotes.
2. The authors should provide GPC traces of the chain extension of PVP-CTA with St/ITA.
3. The authors should provide 1H NMR spectrum of the PVP-CTA.
4. How do the authors confirm the polymerization conversion?
5. How do the authors confirm the particle size and Zeta-potential?
6. The authors should double-check the structure of the PVP-b-P (St/ITA).
7. The authors should check the format of the references.
Reviewer 3 Report
The aim of the report is to optimize the hydrolysis resistance and dispersion of aluminium nitride (AIN) by its surface modification with PVP-3 b-P(St-alt-ITA) copolymer. The paper has its significance and can be published in Molecules taking into account the following comments:
Comments:
(1) The last paragraph of the Introduction seems to very comprehensive and is more attractive as an abstract since the main results are given here. In my opinion, the authors can reconsider the Abstract and the last paragraph of the Introduction.
(2) p. 3, line 120: Why Figure 11 is given here? The synthesis of the copolymer has to be Figure 1?
(3) p. 6, line 203: 1H NMR instead of the written H NMR.
(4) Figure 5, p. 7 – the abbreviations R-AIN and P-AIN are not given The same for Figure 6 on the same page.
Round 2
Reviewer 2 Report
The manuscript is revised well and it can be accepted now.